# CONTEXT-AWARE SELF-TRAINING FRAMEWORK FOR CELL TYPE ANNOTATION USING MARKER GENES

## ABSTRACT

Single-cell annotation is a fundamental task in the analysis of single-cell data, and one promising research direction relies on the marker gene information accumulated in biology. Recently, self-training strategies have been introduced into the field, which significantly improve the annotation accuracy by iteratively optimizing the model. However, existing methods have not yet systematically explored how to construct self-training frameworks that are more applicable to single-cell data. To this end, we propose the context-aware self-training model CSSTA. First, the contextual information of marker genes is introduced to enhance the compatibility of marker genes with different single-cell datasets to generate high-quality pseudo-labels. Second, high- and low-confidence pseudo-labels recognition and supervision strategies more applicable to single-cell data are designed that can better guide the optimization of the model. Finally, the insight of the single-cell foundation model on cell-cell association information is introduced by GNN. Experiments demonstrate that the introduction of marker gene contextual information significantly improves the ability to recognize cell-cell type associations with heuristic-based strategies. Benchmark experiments show that CSSTA significantly outperforms state-of-the-art methods. Notably, we demonstrate the potential of CSSTA for hierarchical cellular annotation by extending it to hierarchies.

## 1 INTRODUCTION

Emerging single-cell RNA sequencing (scRNA-seq) technologies have allowed us to monitor biological systems at a much higher resolution. The identification of distinct cell types within complex tissues contributes to a deeper understanding of their roles in biological processes (Liu et al., 2021). Over the years, the field has established an extensive repertoire of cell-specific features, including widely validated marker genes that facilitate precise cell classification in tissue samples (Franzén et al., 2019). These marker-based approaches have become a cornerstone of cell type annotation (Ianevski et al., 2022; Chen et al., 2024; Hou & Ji, 2024; Busarello et al., 2025).

Traditional marker-based methods mainly use heuristic strategies to assess the association between cells/cell clusters and cell types (Shao et al., 2020; Mikolajewicz et al., 2022). They mimic manual annotation by quantifying the overlap between cell/cluster-specific genes and canonical marker genes through statistical measures (Shao et al., 2020). However, recent studies have revealed significant noise in the cell-type association scores derived from such heuristic approaches (Chen et al., 2024). To overcome these limitations, advanced computational frameworks (Chen et al., 2024; Amini et al., 2025) integrate self-training mechanisms from machine learning to denoise association scores and enhance cell classification performance. As illustrated in Figure 1, the self-training architecture for single-cell annotation operates through three synergistic modules: pseudo-label generation based on heuristic-based association scores (Figure 1a), iterative refinement of classifiers and pseudo-labels (Figure 1b), and cell prediction by neural classifiers (Figure 1c). While existing approaches are mainly preliminary applications of the self-training paradigm, this work systematically addresses three fundamental challenges in the self-training workflow to develop self-training models that are more applicable to single-cell data.

First, the effectiveness of self-training is thought to benefit from the quality of the pseudo-labels, i.e., the performance of the heuristic association scoring strategy (Chen et al., 2024). scCATCH (Shao et al., 2020) assesses the degree of overlap of cell cluster-specific genes with cell type marker genes by counting strategies (Figure 2a). Ianevski et al. (2022) find that rare marker genes appearing

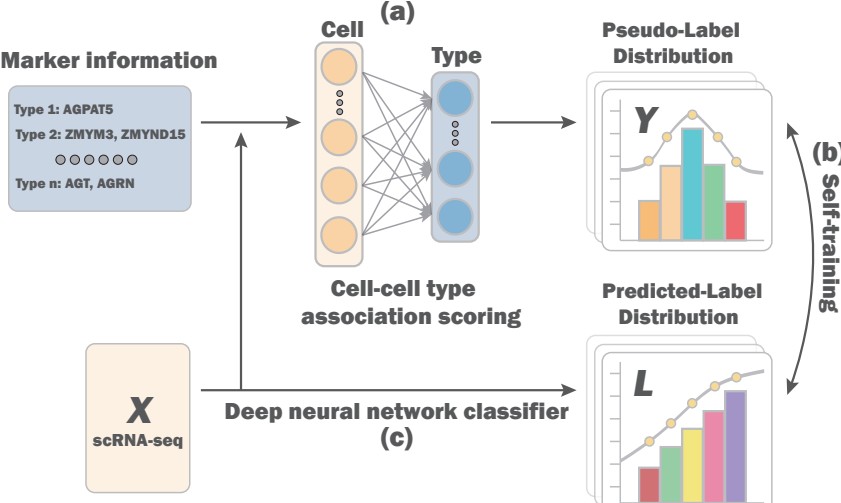

Figure 1: Schematic diagram of the self-training-based single-cell annotation framework. (a) Pseudo-label generation process. (b) Self-training process for supervising downstream classifiers using pseudo-labels. (c) Network architecture of the downstream classifiers.

in fewer cell types may be more discriminatory, and therefore propose a cell type-specific score (Figure 2b). However, these strategies ignore the effects of experimental noise and high rates of loss of single-cell data on the compatibility between marker genes and specific single-cell datasets. When these genes lose their discriminatory power due to experimental noise or high dropout rates on a particular dataset, this scheme may introduce errors rather than improve the accuracy of the annotation (Figure 2c). Therefore, we propose that just as human language is context-dependent (Devlin, 2018; Liang et al., 2023; Naveed et al., 2025), the effectiveness of marker genes is inherently contingent upon the intrinsic characteristics of the processed data.

Second, self-training methods are commonly used to filter out possible noise in pseudo-labels, thereby enhancing model performance. However, most existing self-training methods adopt global thresholds to select high-confidence samples as pseudo-labels (Chen et al., 2024; Yoon et al., 2024). This simple strategy can be problematic for single-cell data, which often suffers from the problem of category imbalance and varying levels of classification difficulty across cell types. Since high-confidence pseudo-labels are typically associated with easily classified samples, an over-reliance on them may impair the model's ability to generalize to more challenging cases. As a result, these methods often overlook reliable predictions for hard-to-classify categories.

Third, a number of methods have made progress by integrating clustering algorithms. MarkerCount (Kim et al., 2022) first evaluates the association between cells and cell types using a counting strategy, and then through principal component analysis and Gaussian mixture modeling to identify cell clusters, and finally corrected the initial association after eliminating uncertain cells. Based on this, HiCAT (Lee et al., 2023) further introduced hierarchical marker gene information to identify major types, minor types, and subsets hierarchically. Although these methods divide the cell classification process into two phases, clustering and assignment, and the marker information of cell types cannot effectively guide the unsupervised cell clustering optimization, they have shown promising performance in previous evaluations, indicating the importance of cell association information, which has been neglected by previous self-training based single-cell annotation methods.

To address the above challenges, we propose a Context-aware Self-training model for Single-cell Type Annotation (CSSTA)[1], as shown in Figure 3. To begin with, we propose the Contextualized Association Scoring (CAS) strategy. This strategy innovatively combines cell-type-specific scoring (Ianevski et al., 2022) with contextualized information of marker genes to effectively generate high-quality pseudo-labels. Subsequently, existing methods rely excessively on high-confidence pseudo-labels during self-training iterations, which may lead to biased adaptation to familiar samples

---

[1]The source code of our CSSTA is available at `https://anonymous.4open.science/r/CSSTA-FDEA`.

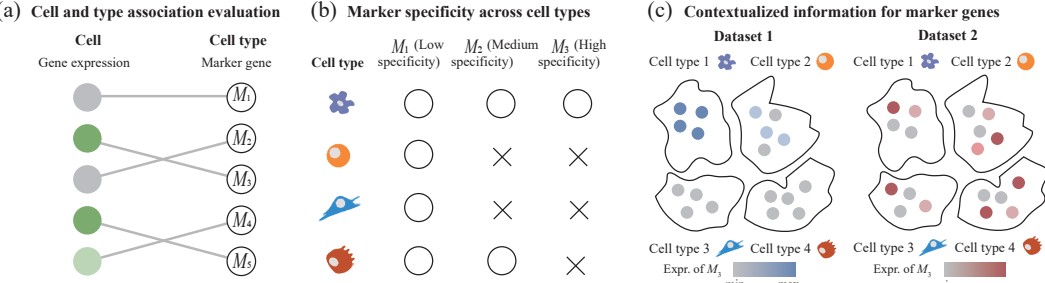

Figure 2: Schematic representation of the key ideas of different association scoring strategies. (a) Cell and cell type association evaluation is based on counting the expression values of all marker genes on the cell for a given cell type. (b) Marker gene specificity assesses whether marker genes are present in a small number of types, e.g., $M_3$ that occurs only in cell type 1 has high specificity. Circles indicate presence in this cell type and vice versa forks. (c) The validity of a marker is affected by the contextualized information of the dataset in which they are located. For instance, $M_3$ has low discriminatory power on dataset 2. Dots represent $M_3$ expression values on cells.

or specific cell types. Therefore, we introduce metric learning to distinguish high-confidence and low-confidence pseudo-labels by calculating the distance between cell embeddings and cell type prototypes, and implement differentiated supervisory strategies for pseudo-labels with different confidence levels to better guide model optimization. Lastly, we integrate cell network topology knowledge. We utilize a pre-trained single-cell foundation model to obtain high-quality cell embeddings and construct a cell association network accordingly. Learning cell association information through Graph Neural Network (GNN) allows CSSTA to effectively capture the spatial relationships between cells, which helps to improve the annotation accuracy. Experimental results show that CSSTA significantly outperforms state-of-the-art methods on several benchmark datasets. Notably, CSSTA can be naturally extended to hierarchical cell annotation tasks and has demonstrated excellent performance on example datasets, providing a new solution for hierarchical cell classification.

## 2 METHODOLOGY

This section details our proposed CSSTA. The development of this model focuses on 3 challenges of current self-training based single-cell annotation frameworks. **RQ1:** How to quantify and utilize marker contextualized information to enhance the quality of pseudo-labels? **RQ2:** How to optimize the process of guiding models with pseudo-labels? **RQ3:** How to integrate the knowledge of the cell-cell topology to enhance annotation performance? Below, we first present the details of the relevant techniques, and then describe the framework of CSSTA as shown in Figure 3. Finally, we provide details of model inference and training.

### 2.1 MEASURING THE RELEVANCE BETWEEN CELLS AND CELL TYPES

How to assess association scores between cells and cell types based on marker genes has been extensively studied. Here, we introduce six commonly used association scoring strategies, two of which (i.e., Count and Cell-type-specific) are described below and the others in Appendix A. Then we present our strategy CAS. Specifically, single-cell expression data is denoted as $X \in \mathbb{R}^{n \times g}$, where $n$ and $g$ are the number of cells and genes, respectively. $x_i$ denotes the $i_{\text{th}}$ cell in $X$. Assuming that there are $\mathcal{T}$ known cell types, each type $\mathcal{T}_j$ has a list of marker genes $\{m_j^k\}, k = 1 \ldots n_j$. The different scoring strategies are described below.

(i) **Count-based correlation scoring (Count):** The relevance between cell $x_i$ and type $\mathcal{T}_j$ is assessed by checking whether the marker genes in $\mathcal{T}_j$ are expressed in $x_i$. It can be described as follows:

$$R_{ij} = \sum_{k=1}^{n_j} \mathbf{I}(x_i \left[m_j^k\right] > 0),$$ (1)

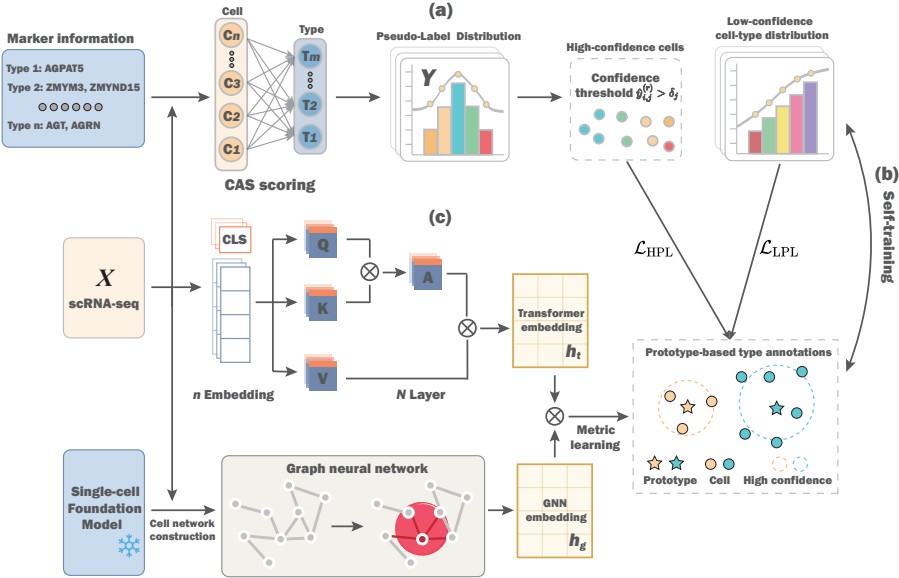

Figure 3: Network architecture of CSSTA. (a) Pseudo-labels generated by CAS scoring. (b) High-confidence and low-confidence pseudo-labels are identified based on cell type prototypes and cell embedding distances, and the classifier optimization is guided in using the discretization loss function. (c) Integration of cell network topology knowledge in single-cell foundation models via GNN.

where $\mathbf{I}(x_i\left[m_j^k\right] > 0)$ denotes whether the $k_{\text{th}}$ marker gene of the $j_{\text{th}}$ cell type $\mathcal{T}_j$ is expressed in $x_i$. If $x_i\left[m_j^k\right] > 0$, then $\mathbf{I}(x_i\left[m_j^k\right] > 0) = 1$, and vice versa for 0.

(ii) **Relevance scoring based on marker specificity for cell types (Cell-type-specific):** Since the same gene may appear in the list of markers corresponding to multiple cell types, Ianevski et al. (2022) propose a Cell-type-specific score, which quantifies the specificity of a marker, with higher specificity indicating that the marker gene occurs less frequently. Specifically, given a marker pool $M$ that contains marker genes from all cell types, the score $S_j$ for marker $M_j$ is defined as follows:

$$S_j = 1 - \frac{|M_j| - |M_{\min}|}{|M_{\max}| - |M_{\min}|}, \tag{2}$$

where $|M_j|$ denotes the number of marker genes $M_j$, and $|M_{\min}| = \min_{M_k \in M} |M_k|$ and $|M_{\max}| = \max_{M_k \in M} |M_k|$ are the minimum and maximum number of marker genes. The expression of the genes in each cell is then multiplied by the $S$ scores to derive the cell-cell type score $R_{ij}$, i.e.,

$$\boldsymbol{X'} = \left(\left(Z\left(\boldsymbol{X}^T\right)\right)^T \subseteq M\right) \cdot S, \quad R_{ij} = \frac{1}{\sqrt{n_j}} \sum_{k=1}^{n_j} x'_i\left[m_j^k\right], \tag{3}$$

where $x'_i$ means the $i_{\text{th}}$ cell in $\boldsymbol{X'} \in \mathbb{R}^{n \times |M|}$ and $Z(\cdot)$ denotes the z-score-transformation.

(iii) **Relevance scoring based on marker specificity and contextualization (CAS):** Previous studies (Wang et al., 2019; Pullin & McCarthy, 2024) mention that marked genes are expected to show large expression differences between cell types, i.e., it should be up-regulated in relevant cell types, while showing low expression in the other types. To quantify the above property of markers considering the specificity and contextualized information in a given dataset, we define 3 metrics following the understanding of biologists:

∘ **Unique** Marker genes are unique, which means they should appear less frequently. Given that $f_{\boldsymbol{X}, m_j^k} = \sum_{i=1}^n \mathbf{I}(x_i\left[m_j^k\right] > 0)$ denotes the occurrence frequency of the marker gene $m_j^k$, the uniqueness score for $m_j^k$ is defined as follows:

$$U\left(m_j^k\right) = \log\left(\frac{n}{f_{\boldsymbol{X}, m_j^k}}\right). \tag{4}$$

○ **Co-occurring** Given the marker gene $m_j^k$, the conditional probability $P\left(\mathcal{T}_j \mid m_j^k\right)$ of a cell belonging to the cell type $\mathcal{T}_j$ should be high, which we call the co-occurrence of the marker with its corresponding cell type. With $f_{\mathcal{T}_j, m_j^k} = \sum_{x_i \in \mathcal{T}_j} \mathbf{I}(x_i\left[m_j^k\right] > 0)$, the co-occurring score is calculated as follows:

$$C\left(\mathcal{T}_j, m_j^k\right) = P\left(\mathcal{T}_j \mid m_j^k\right) = \frac{f_{\mathcal{T}_j, m_j^k}}{f_{\boldsymbol{X}, m_j^k}}. \tag{5}$$

○ **Frequent** Ideally, marker genes occur as frequently as possible in their corresponding cell types, which is defined as follows:

$$F\left(\mathcal{T}_j, m_j^k\right) = \text{relu}\left(\frac{f_{\mathcal{T}_j}\left(m_j^k\right)}{f_{\mathcal{T}_j}}\right), \tag{6}$$

where $f_{\mathcal{T}_j} = \sum_{i=1}^n \mathbf{I}(x_i \in \mathcal{T}_j)$ and $f_{\mathcal{T}_j}\left(m_j^k\right) = \sum_{x_i \in \mathcal{T}_j} x_i\left[m_j^k\right]$ denote the number of cells belonging to the type $\mathcal{T}_j$ and the frequency of $m_j^k$ in them, respectively. Since the average frequency is unbounded, we scale it with the relu function. Finally, we combine these three measures using geometric averaging to derive contextualized scores:

$$O\left(\mathcal{T}_j, m_j^k\right) = \left(U\left(m_j^k\right) \times C\left(\mathcal{T}_j, m_j^k\right) \times F\left(\mathcal{T}_j, (m_j^k)\right)\right)^{1/3}. \tag{7}$$

Next, we utilize marker's score $O$ and the Cell-type-specific score, resulting in a CAS score with better compatibility. Specifically, we first obtain the initial pseudo-label by the Cell-type-specific score and use this as a basis for calculating $O$ score (Eqs. (4-7)). Then, the $O$ score is combined with the Cell-type-specific score to obtain the final CAS score, as shown in the following equation:

$$\boldsymbol{X}'' = \boldsymbol{X}' \cdot N(O), \quad R_{ij} = \frac{1}{\sqrt{n_j}} \sum_{k-1}^{n_j} x_i''\left[m_j^k\right], \tag{8}$$

where $x_i''$ is the $i_{\text{th}}$ cell in $\boldsymbol{X}''$ and $N(\cdot)$ denotes the normalization-transformation.

## 2.2 GUIDING MODEL OPTIMIZATION THROUGH PSEUDO-LABELS

This section describes how our CSSTA uses pseudo-labels to guide model optimization.

(i) **Identification of high and low-confidence pseudo-labels:** To address the prevalent noise problem in pseudo-labels, existing studies such as Li et al. (2022) and Chen et al. (2024) employ global thresholding to filter low-confidence predictions, but this approach is difficult to adapt to the inherent type imbalance characteristics in single-cell data. To this end, we propose a type-adaptive thresholding scheme that sets an exclusive threshold $\delta_j$ for the $j_{\text{th}}$ cell type. The specific implementation process is as follows: in the initialization stage of the $r_{\text{th}}$ round of iteration, generate the pseudo-label $\tilde{y}^{(r)} = \arg\max_j \hat{y}_j^{(r-1)}$ based on the current model prediction probability distribution $\hat{y}^{(r-1)}$; and subsequently, for each cell type $T_j$, screen all the cell types that have been labeled with this type sample set $X[T_j] = \{x_i | \tilde{y}_i^{(r)} = j\}$; ultimately, based on the predictive probability distribution, the samples of each type are sorted by confidence, and the top $\epsilon\%$ (corresponding to the threshold $\delta_j$) are taken as the high-confidence sample set $X^H[T_j]$. Except for the high-confidence samples of each type, the rest are categorized into the low-confidence sample set $X^L[T_j]$, which is defined as follows:

$$X^H\left[T_j\right] = \left\{x_i \mid x_i \in X\left[T_j\right], \hat{y}_{i,j}^{(r)} > \delta_j\right\}, \tag{9}$$

$$X^H = \bigcup_{j=1}^l X^H\left[T_j\right], X^L = X\left[T_j\right] - X^H \tag{10}$$

(ii) **Loss function for low-confidence pseudo-labels:** For the design of the loss function for low-confidence pseudo-labels, we consider in depth the special properties of such samples and their potential impact on model training. Low-confidence samples usually correspond to difficult instances near the classification boundary, and their prediction reliability is relatively low, but completely ignoring these samples will cause the model to be overly biased towards simple instances, thus impairing the discriminative ability for complex samples. To address this challenge, we design a loss function mechanism based on KL dispersion to optimize Low-confidence Pseudo Labels (LPL)

are utilized efficiently. Specifically, this loss function is mathematically expressed as the KL scatter between the predictive distribution and the pseudo-label distribution by the following equation:

$$\mathcal{L}_{\text{LPL}}(\tilde{\mathbf{Y}}^{(r)}, \hat{\mathbf{Y}}^{(r+1)}) = \sum_{x_i^L \in X^L} \sum_{j=1}^{l} \tilde{y}_{ij}^{(r)} \log \frac{\tilde{y}_{ij}^{(r)}}{\hat{y}_{ij}^{(r+1)}}. \tag{11}$$

We employ soft supervision based on probability distributions, which can more accurately characterize the uncertainty in the cell type boundary regions by preserving the relative similarity relationship between cell and type prototypes.

(iii) **Loss function for high confidence pseudo labels:** Unlike LPL, HPL is widely recognized as a reliable supervised signal in traditional self-training methods. To more fully utilize these high-quality samples, we introduce an additional cross-entropy loss function on top of the KL divergence loss to further distance the high confidence samples from the corresponding types of prototypes in the embedding space, thus strengthening the model's ability to discriminate explicit samples. The HPL loss function is specifically defined as follows.

$$\mathcal{L}_{\text{HPL}}(\tilde{\mathbf{Y}}^{(r)}, \hat{\mathbf{Y}}^{(r+1)}) = \sum_{x_i^H \in X^H} \sum_{j=1}^{l} \left[ \hat{y}_{ij}^{(r+1)} \log \frac{\hat{y}_{ij}^{(r+1)}}{\tilde{y}_{ij}^{(r)}} - \mathbf{1}\left(\tilde{y}_i^{(r)} = j\right) \cdot \log \hat{y}_{ij}^{(r+1)} \right]. \tag{12}$$

### 2.3 DUAL-VIEW NETWORK ARCHITECTURE

To introduce comprehensive cell-cell association information, we utilize a single-cell foundation model Geneformer (Theodoris et al., 2023) and design a dual-view network architecture consisting of two branches, GNN and Transformer. The Transformer is used to extract gene-level information, including gene expression and gene-gene interactions. The GNN is used to extract cell-level information, including the associations between cells. We consider cell representations obtained from the single-cell foundation model as the coordinates of the global cell space, and use them as the basis for constructing a cell-cell network $G$, which is used as an input to the GNN. Detailed descriptions of the branches are described in the Appendix B. We input the single-cell data into the GNN branch $f_g$ and the Transformer branch $f_t$, and obtain the corresponding cell embeddings $h_g$ and $h_t$, respectively.

$$h_g = f_g(\mathbf{X}, G), \quad h_t = f_t(\mathbf{X}), \quad h_f = \text{columnbind}(h_g, h_t). \tag{13}$$

In contrast to approaches that use feedforward neural networks as classifiers, we transform the cell classification process into a metric learning problem in the embedding space. Specifically, we map cell embeddings $h_f^i$ and learnable cell type prototypes $\mu_j$ into the same feature space and optimize the distance between them by pseudo-label supervision. Specifically, for the $i$-th cell with embedded feature $\mu_j$ and the $j$-th cell type prototype with embedded feature $h_f^i$, we compute their distances via

$$\tilde{y}_{i,j} = \frac{\exp\left(\text{sim}\left(h_f^i, \mu_j\right)/\tau\right)}{\sum_{j=1}^{l} \exp\left(\text{sim}\left(h_f^i, \mu_j\right)/\tau\right)}, \tag{14}$$

where $\text{sim}(\cdot, \cdot)$ denotes the cosine similarity and $\tau$ is the temperature. More details on model training can be found in Appendix C.

## 3 EXPERIMENTS

### 3.1 EXPERIMENTAL SETUP

To demonstrate the effectiveness of CSSTA in the task of cell type annotation, we collected 8 scRNA-seq datasets with manual annotations including different species, tissues, and scales for our experiments. The dataset details are given in Appendix D. To better evaluate the performance of CSSTA, we use 7 state-of-the-art single-cell annotation methods as benchmark models, including Garnett (Pliner et al., 2019), SCINA (Zhang et al., 2019), scSorter (Guo & Li, 2021), scType (Ianevski et al., 2022), MarkerCount (Kim et al., 2022), Geneformer (Theodoris et al., 2023), and sICTA (Chen et al., 2024). The details of the implementation of the CSSTA and benchmarking models are given in Appendix E.

### 3.2 COMPARING DIFFERENT CORRELATION SCORING STRATEGIES

Table 1: Performance of different methods, where **Avg** represents the average of the performance over all single-cell datasets, **Bold** denotes the optimal result, and underline denotes the second best result. **Std** denotes the standard deviation of all single-cell datasets, and its smaller the better.

| Model | Avg↑ | Std↓ | Muraro | Stoeckius | Zheng | Tirosh | Puram | Zeisel | Dominguez-Conde | Yoshida |
|---|---|---|---|---|---|---|---|---|---|---|
| **Macro-F1** | | | | | | | | | | |
| scSorter | 0.483 | 0.287 | 0.936 | 0.419 | 0.204 | 0.449 | 0.092 | 0.839 | 0.469 | 0.481 |
| Garnett | 0.376 | 0.181 | 0.471 | 0.421 | 0.092 | 0.349 | 0.294 | 0.786 | 0.279 | 0.470 |
| SCINA | 0.568 | 0.165 | 0.842 | 0.754 | 0.479 | 0.774 | 0.340 | 0.604 | 0.448 | 0.391 |
| scType | 0.584 | 0.132 | 0.884 | 0.639 | 0.479 | 0.625 | 0.545 | 0.624 | 0.433 | 0.440 |
| MarkerCount | 0.680 | 0.138 | 0.806 | 0.821 | 0.763 | 0.808 | 0.788 | 0.760 | 0.440 | 0.502 |
| Geneformer | 0.682 | **0.110** | 0.609 | 0.803 | 0.772 | 0.730 | 0.795 | 0.708 | 0.530 | 0.511 |
| sICTA | 0.785 | 0.133 | 0.954 | 0.926 | 0.753 | 0.810 | 0.792 | 0.837 | 0.575 | 0.537 |
| CSSTA | **0.832** | 0.129 | **0.964** | **0.932** | **0.780** | **0.863** | **0.902** | **0.952** | **0.672** | **0.593** |
| **Micro-F1** | | | | | | | | | | |
| scSorter | 0.644 | 0.234 | 0.949 | 0.720 | 0.722 | 0.809 | 0.384 | 0.874 | 0.551 | 0.505 |
| Garnett | 0.539 | 0.171 | 0.631 | 0.709 | 0.383 | 0.569 | 0.761 | 0.821 | 0.403 | 0.493 |
| SCINA | 0.620 | 0.189 | 0.900 | 0.895 | 0.798 | 0.623 | 0.421 | 0.564 | 0.531 | 0.426 |
| scType | 0.657 | 0.133 | 0.921 | 0.790 | 0.543 | 0.568 | 0.712 | 0.649 | 0.517 | 0.459 |
| MarkerCount | 0.754 | 0.148 | 0.795 | 0.942 | **0.893** | 0.754 | 0.937 | 0.843 | 0.525 | 0.664 |
| Geneformer | 0.721 | 0.131 | 0.729 | 0.924 | 0.877 | 0.682 | 0.738 | 0.746 | 0.547 | 0.522 |
| sICTA | 0.835 | 0.122 | 0.967 | 0.947 | 0.871 | 0.698 | 0.967 | 0.886 | 0.712 | 0.600 |
| CSSTA | **0.895** | **0.107** | **0.975** | **0.971** | 0.886 | **0.948** | **0.973** | **0.964** | **0.777** | **0.667** |

To find the optimal heuristic correlation assessment strategies to generate higher quality initial pseudo-labels, this section evaluates the performance of 7 correlation assessment strategies (Count, Cos, LR-label, LR-marker, Pseudo-cell, Cell-type-specific, and CAS) on 8 single-cell datasets using Macro-Precision, Macro-Recall, Macro-F1 and Micro-F1 metrics. The results are shown in Figure 4. Among all association assessment strategies, the Cell-type-specific strategy achieved the second best performance, compared to the third ranked Count, which effectively utilized the specificity between marker genes and cell types. CAS achieved the overall optimal performance, with an average improvement of 6.2% on two comprehensive evaluation

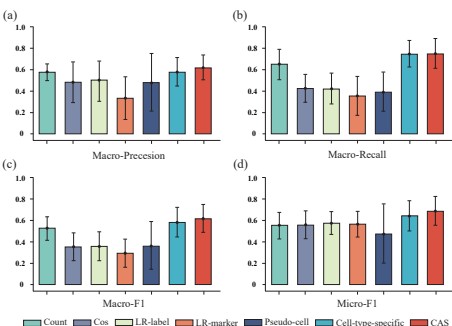

Figure 4: Average performance of different association scoring strategies.

metrics (Macro-F1 and Micro-F1) compared to Cell-type-specific, demonstrating the effectiveness of incorporating marker gene contextual information. The pseudo-cell strategy obtains overall lower and unstable performance, which may be due to the fact that on some datasets its generated cells differ significantly from the real data distribution In summary, the above experiments show that by introducing contextual information of the marker genes, CAS can better adapt to different single-cell datasets and improve the accuracy of recognizing cell type associations. Therefore, the proposed CAS is used to generate the initial pseudo-labels.

### 3.3 PROVIDING HIGH QUALITY CELL TYPE ANNOTATIONS

In this section, we benchmark CSSTA and compare it to 7 benchmark methods for single-cell annotation. The results (Table 1 and 2) show that CSSTA significantly outperforms other methods, with 6.0% higher Macro-F1, and 7.2% higher Micro-F1 than the previous self-training based sICTA, which proves the effectiveness of our optimization of the self-training framework. Compared to scType, CSSTA improved by 35.5% on average on all four metrics, suggesting that the context-aware self-training strategy and the powerful nonlinear fitting ability of the dual-view structure greatly improve the model's ability to capture potential dependencies between cells and types. Furthermore, we find that scSorter's performance varies significantly across datasets, as it tends to predict cells as categories with higher sample sizes when faced with unbalanced datasets. In contrast, the standard deviation of CSSTA on different datasets shows greater robustness. We then examined the changes in different metrics of CSSTA during self-training, and the results on all single-cell data consistently showed that the performance of CSSTA gradually improved until convergence during self-training, which further demonstrated the robustness of the model (Figures 5 and 8).

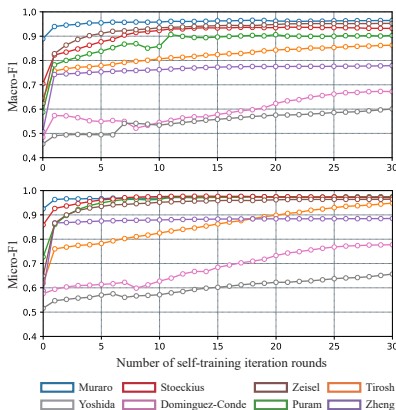

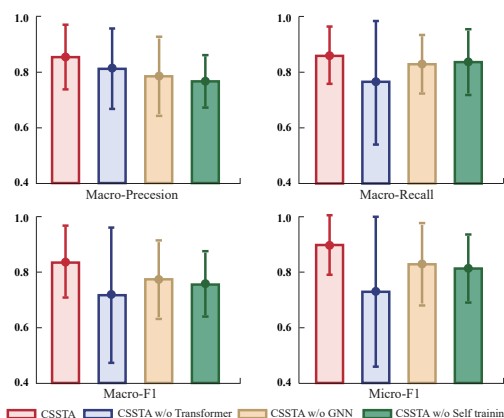

Figure 5: Plot of performance variation of CSSTA in self-training.

Figure 6: Impact of different branches on model performance.

Next, we further refine our evaluation of model performance. Since the number of different cell types in single-cell datasets is not balanced, for instance, the number of Tcm/Naive helper T cells in the Yoshida dataset is 56 times that of Memory B cells, and the number of T cells in the Tirosh dataset is 45. 6 times. Therefore, we evaluate the prediction accuracy of CSSTA and the two best benchmark models (MarkerCount and sICTA) on the rarest and highest percentage of cell types. The results (Tables 3 and 4) show that CSSTA achieves optimal performance on both cell types. Besides, CSSTA accurately identifies cell types that are not recognized by MarkerCount and sICTA, such as the rarest cell types in the Puram dataset in Table 3.

### 3.4 REVEALING THE EFFECTIVENESS OF DIFFERENT BRANCHES

To explore the impact of different branches in the model on performance, we ablated the framework of CSSTA and evaluated the ablated model on all datasets. Specifically, we evaluated the following scenarios: (i) ablation of the Transformer branch (CSSTA w/o Transformer); (ii) ablation of the GNN branch (CSSTA w/o GNN); (iii) eliminating the self-training strategy (CSSTA w/o self-training). Figure 6 shows the average performance of CSSTA in each ablation over multiple datasets.

The results show that the Transformer branch has the greatest impact on the results, and when the Transformer branch is removed, the model exhibits lower stability, due to the fact that this branch contains the essential cell expression data. The self-training process also has a significant impact on the performance, which is in line with the findings of previous work (Chen et al., 2024), further demonstrating the effectiveness of the self-training process. Furthermore, the average performance of CSSTA on Macro-F1 and Micro-F1 is improved by 7.8% and 8.4%, respectively, compared to CSSTA w/o GNN, which proves the importance of introducing cell network information.

### 3.5 EXPLORING STRATEGIES FOR PSEUDO-LABEL GENERATION AND SUPERVISION

To explore the impact of pseudo-label generation and supervisory strategies on model performance, we performed a comprehensive ablation. Specifically, we compare strategies that do not focus on low confidence pseudo-labels (CSSTA w/o LPL), that use all cells for training (CSSTA w AC) or that distinguish between high and low-confidence pseudo-labels by a global threshold (CSSTA w GT). To ensure consistency in the experimental setup, the global threshold was set to the $\epsilon$%, i.e. the $\epsilon$% of samples with the highest probability scores are considered as high-confidence pseudo-labels. In addition, the performance of Cell-type-specific substitution of CAS when utilized to generate pseudo-labels (CSSTA w/o CAS) is also compared.

As shown in Table 5, despite achieving comparable performance with the ablation model on some datasets, CSSTA significantly outperforms the ablation model in terms of overall performance. When ignoring low-confidence pseudo-labels or utilizing Cell-type-specific to generate pseudo-labels, the average performance of the model on the four metrics decreases by 10.0% and 11.5%, respectively, which demonstrates the importance of these two components for CSSTA. Moreover, we observe

that although CSSTA w GT and CSSTA w AC can achieve good performance on specific datasets, their overall classification accuracy is poor, which suggests that they are sensitive to different data distributions and do not have the ability to generalize.

### 3.6 VERIFYING THE ROBUSTNESS OF CSSTA FROM DIVERSE PERSPECTIVES

To comprehensively evaluate the stability and generalization capability of the CSSTA model in practical applications, we validated its robustness using two representative datasets, Tirosh and Zheng, which differ in scale and sequencing technology. The evaluation focused on three aspects: (i) robustness to noisy marker gene lists; (ii) sensitivity to key hyperparameter choices; and (iii) scalability when integrated with different single-cell foundation models (see Appendix F for details).

First, by randomly masking marker genes (10%-30%), we find that both the CAS module and the full CSSTA model demonstrate good resilience (Tables 6 and 7), indicating insensitivity to marker gene noise. Second, sensitivity analysis shows that the model's performance remained stable against variations in key hyperparameters (confidence threshold $\epsilon\%$ and temperature $\tau$) within a reasonable range (Tables 8 and 9). Finally, by integrating CSSTA with different foundation models like scGPT (Cui et al., 2024) and CellPLM (Wen et al., 2023), it consistently and significantly outperformed the baselines (Table 10), demonstrating the framework's generality. These experiments collectively indicate that CSSTA exhibits high robustness.

### 3.7 EXTENDING CSSTA TO HIERARCHICAL CELL CLASSIFICATION TASKS

With the deeper study of the single-cell field, researchers begin to define different levels of cell types (Miller et al., 2020; Kim et al., 2022), such as major and minor types, and organize them into hierarchical structures, which allows for cells to be understood and analyzed in a much finer-grained way. Therefore, it is an important challenge to utilize the structural knowledge of cell types so that cells can be accurately assigned to cell types at different levels. Here, inspired by hierarchical text categorization methods (Meng et al., 2019; Ji et al., 2023), CSSTA is extended to hierarchical cell annotation (h_CSSTA), as shown in Figure 7. See Appendix G for a more detailed description.

To assess the ability of h_CSSTA to perform hierarchical classification of cell types, we perform experiments on the example dataset provided by Lee et al. (2023), where the cell types are categorized into five major and seven minor types. We use the hierarchical cell annotation method HiCAT (Lee et al., 2023), as a baseline.

The results (Table 11) show that h_CSSTA significantly outperforms the latest baseline model. Compared with HiCAT, the performance of h_CSSTA at major and minor types is improved by 19.7% and 11.3%, respectively, indicating that h_CSSTA benefits from the explicit modeling the structural information of cell types. In addition, we observe that h_CSSTA improves its performance on major cell types compared to CSSTA, which suggests that h_CSSTA is capable of correcting the errors of the upper layers by training the lower layers, validating the strong potential of h_CSSTA.

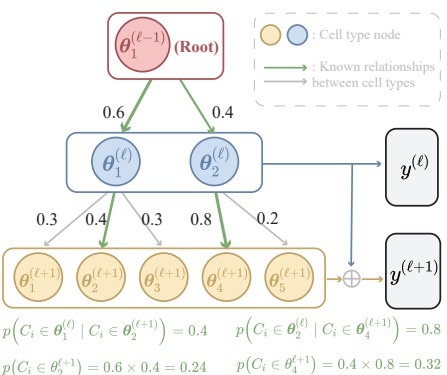

Figure 7: Schematic diagram of h_CSSTA.

## 4 CONCLUSION

We propose CSSTA, a context-aware self-training framework tailored for single-cell data. CSSTA quantifies the contextual information of marker genes to enhance dataset compatibility, employs a differentiated supervision strategy adapted to single-cell data characteristics, and incorporates external cell topology to boost performance. Experiments demonstrate that our model achieves state-of-the-art performance in cell type annotation. Extensive ablation and robustness analyses validate the model's effectiveness. Finally, we extend CSSTA to hierarchical cell annotation and validate its outstanding performance on example datasets, providing a highly promising approach for addressing cell hierarchy classification tasks.

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

## A  RELEVANCE SCORING STRATEGY

(i) **Relevance scoring based on external knowledge and cosine similarity (Cos):** Here we introduce the gene embedding $\mathbf{E_g}$ generated by previous work (Du et al., 2019). Based on the expression data of cells and $\mathbf{E_g}$, the embedding representation of each cell ($c_i^{emb}$) is obtained by weighted averaging. And the embedding representation of each cell type ($\mathcal{T}_j^{emb}$) is the average of its marker gene embeddings.

$$c_i^{emb} = \frac{x_i \cdot \mathbf{E_g}}{\text{sum}(x_i)}, \quad \mathcal{T}_j^{emb} = \frac{\sum_{k=1}^{n_j} e_j^k}{n_j} \tag{15}$$

where $e_j^k$ is the embedding of its corresponding marker gene $m_j^k$. Next, the cosine similarity between cell embeddings and type embeddings is used as the association score between them.

$$R_{ij} = cosine(c_i^{emb}, \mathcal{T}_j^{emb}) \tag{16}$$

(ii) **Relevance scoring based on external knowledge and logistic regression (LR-label & LR-marker):** Instead of cosine similarity, we can employ logistic regression classifiers trained on external knowledge to infer associations between cells and candidate types. In the LR-label strategy, we consider the cell type and its corresponding marker as a cell containing multiple genes. Therefore, we obtain the embedding representation of this sample by averaging the marker gene embeddings and use it to train the classifier (one cell sample per class). Oppositely, in the LR-marker strategy, each marker gene contained in a cell type is considered to be a representation of that cell type, and thus each marker gene embedding is used as a training sample (multiple cell samples per class).

(iii) **Pseudo-cell-based correlation scoring (Pseudo-cell):** This strategy is adapt from the pseudo-document generation method proposed by Meng et al. (2018). We map cells and genes into the same embedding space, where each cell type is modelled as a high-dimensional spherical distribution. Then pseudo-cells are generated by sampling from the type distributions. Next, pseudo-cells and their corresponding type labels are used to train downstream classifiers.

## B ARCHITECTURE OF THE DIFFERENT BRANCHES

(i) **GNN branch:** To exploit the external cellular topology information, we choose the classical GNN architecture (Kipf & Welling, 2016) as the backbone. The GNN fuses representations of neighbouring nodes through an aggregation layer to efficiently capture the structural information of the cellular network. Specifically, for each single-cell dataset, we input its cell expression data into Geneformer to get an embedded representation of each cell. Then, the similarity between each cell pair is quantified by using Cos correlation. Next, the cell-cell network is constructed by selecting the cell pairs with the correlations upper bound of the 1% quantile. Previous methods (Wang et al., 2021; Yu et al., 2022) generally select 10 or 15 nearest neighbor cells to construct the network. We think that as the dataset gets bigger, a fixed number of neighbors might make the network too sparse. So, we use 1% as the threshold to make sure that when we're dealing with larger datasets. It is worth noting that unlike the Geneformer used as a baseline, we froze all the parameters of Geneformer and did not fine-tune them when extracting the cell embeddings.

(ii) **Transformer branch:** Transformer has been widely used in various fields in recent years, and its advanced features in non-linear association modelling help to better exploit single-cell expression data. In addition, its attention mechanism allows explicit construction of gene-gene interactions to better utilize the internal knowledge of cells, hence we have chosen TOSICA (Chen et al., 2023) as the backbone network for this branch. TOSICA is the most recent SOTA single-cell supervised classification model, which stacks multiple multi-head attention layers.

## C TRAINING PROCESS

On the basis of the previous section detailing the methods for distinguishing between pseudo-labels of different confidence levels and their supervisory strategies, we will now illustrate how they supervise the training of the model. The training process is divided into two key stages:

**Pre-training process:** we use the correlation score (CAS) obtained based on the heuristic strategy as the initial pseudo-label of the model, i.e., $\tilde{\mathbf{Y}}^{(0)} = R$. In the initialization phase of the $r$th iteration, we generate the pseudo-label $\tilde{\mathbf{Y}}^{(r)}$ based on the current model's prediction probability $\hat{\mathbf{Y}}^{(r)}$. Considering that the pseudo-labels generated by heuristic methods usually contain more noise (it has been shown that their accuracy is only about 50%, and our experimental results also verify this phenomenon), we adopt a conservative training strategy by utilizing only high-confidence pseudo-labels for the initial training of the model, with the loss function defined:

$$\mathcal{L}_{\text{PreTrain}} = \mathcal{L}_{\text{HPL}}(\tilde{\mathbf{Y}}^{(r)}, \hat{\mathbf{Y}}^{(r+1)}). \tag{17}$$

**Self-training process:** After entering the self-training phase, the pre-trained neural network has already possessed a strong ability to capture nonlinear relationships, and the quality of the generated

pseudo-labels is significantly improved at this time. In this phase, we adopt a more aggressive training strategy, while using high and low-confidence samples for model optimization.

$$\mathcal{L}_{\text{SelfTrain}} = \mathcal{L}_{\text{HPL}}(\tilde{\mathbf{Y}}^{(r)}, \hat{\mathbf{Y}}^{(r+1)}) + \mathcal{L}_{\text{LPL}}(\tilde{\mathbf{Y}}^{(r)}, \hat{\mathbf{Y}}^{(r+1)}). \tag{18}$$

# D  DASASETS

To demonstrate the effectiveness of our CSSTA in the task of cell type annotation, we collected 8 scRNA-seq datasets with manual annotations for our experiments. We used the data provided by Kim et al. (2022). They obtained the marker information from two databases (Panglao (Franzén et al., 2019) and CellMarker (Hu et al., 2023)), and we removed the Genformer pre-training dataset that was used. Additionally, we collected a mouse brain dataset (Zeisel) and two human blood datasets (Dominguez_Conde and Yoshida). The markers for Zeisel were derived from the original paper, while those for Dominguez_Conde and Yoshida were obtained from the CellTypist immune cell atlas (Domínguez-Conde et al., 2022). It is worth noting that for cell expression data, some benchmark models utilize only the expression information of marker genes, and some benchmark models utilize the expression information of non-marker genes. Following the experimental setting of Guo & Li (2021). we set the number of their non-marker genes to the top-2000 highly variable genes. A summary of dataset statistics is shown in Table 12.

# E  HYPERPARAMETER SETTINGS

Our CSSTA is implemented in Python, and the core model is build on the Pytorch (v.1.8.1) framework. The pre-training and self-training phases of the training process took a total of 50 epochs. The optimization of CSSTA is done by AdamW (Loshchilov & Hutter, 2017). In addition, we set all embedding dimensions in CSSTA to 200. For the Transformer architecture, we set its depth to 2 and the number of attention heads to 4. For GNN, we set the number of convolution layers to 2. In addition, for the threshold we empirically chose 20% as the high confidence label, and the temperature $\tau$ was set to 0.05 following previous work (Zhai et al., 2024). The parameter inference method for CSSTA is presented in Algorithm 1.

# F  SUPPLEMENTARY DETAILS FOR ROBUSTNESS EXPERIMENTS

This appendix provides detailed settings for the robustness experiments described in the main text and offers further descriptions of the results.

(i) **Noisy marker gene experiments:** To analyze the robustness of marker gene selection, we randomly masked marker genes with 10%, 20%, and 30% probabilities and then examined the performance of CAS. To ensure the stability of the results, each experiment was repeated five times. We find (Table 6) that CAS maintains relatively stable performance when subjected to 10% and 20% random perturbations. In addition, we further obtain the performance of CSSTA based on these CAS. The results (Table 7) further demonstrate that the model is robust in the face of perturbed marker data.

(ii) **Hyperparameter sensitivity experiments:** We conduct a comprehensive sensitivity analysis of hyperparameters $\epsilon\%$ and $\tau$ to systematically evaluate the robustness of CSSTA. The selection of temperature $\tau$ value followed the work of Zhai et al. (2024). To assess the model's sensitivity to $\tau$, we tried multiple $\tau$ values. The results in Table 8 show that the model maintained good stability for different $\tau$ values around the default value of 0.05. It is only when the $\tau$ value deviates significantly from 0.05 (CSSTA ($\tau = 0.2$)) that the model's performance changes significantly.

In addition, we also explore the impact of different confidence thresholds $\epsilon\%$ on the results. As shown in Table 9, we find that when using higher confidence thresholds such as 10% and 20%, the model still maintained good performance. However, as $\epsilon\%$ further increased, the model performance significantly declined. We think this is mainly because when too many samples are classified as high-confidence samples, it introduces too much noise, which affects the model's optimization direction. This further proves the necessity of the differentiated loss function. In summary, we conclude that the CSSTA can be stabilized within a reasonable range of hyperparameter perturbations.

(iii) **Single-cell foundation model compatibility experiments:** For the foundation models, we further integrate scGPT and CellPLM foundation models, with results shown in Table 10. We observe that CSSTA can be effectively combined with different single-cell foundation models, and even outperforms CSSTA (Genformer) on the Tirosh dataset, further validating the model's scalability.

## G  HIERARCHICAL CSSTA

To extend our CSSTA to h_CSSTA for hierarchical cell type annotation, we start by generating pseudo-labels for cells at each layer via the CAS strategy. Then, for each layer except the leaf layer, we pre-train an internal classifier for each node in it, thus predicting the probability that a cell is assigned to a child node in its lower layer. As shown in Figure 7, each node $\theta_i^{(\ell)}$ corresponds to a classifier $\phi_i^{(\ell)}$. Next, the probability distribution of all categories at level $\ell$ is predicted by ensembling the results from the root classifier to those classifiers at level $(\ell - 1)$. The ensemble method conducts multiplication operation between the output of the parent classifier and the output of the child classifiers which can be explained by the following conditional probability formula:

$$
\begin{aligned}
p\left(x_i \in \mathcal{T}_c^\ell\right) &= p\left(x_i \in \mathcal{T}_c^\ell \cap x_i \in \mathcal{T}_p^{\ell-1}\right) \\
&= p\left(x_i \in \mathcal{T}_c^\ell \mid x_i \in \mathcal{T}_p^{\ell-1}\right) p\left(x_i \in \mathcal{T}_p^{\ell-1}\right),
\end{aligned}
\tag{19}
$$

where $x_i$ is a cell. $\mathcal{T}_c^\ell$ is one of the children type of $\mathcal{T}_p^{\ell-1}$. The final classification prediction is then derived by the multiplication of the results from the root classifier to the classifier at the current level, which enables lower-level classifiers to correct misclassifications at the higher level, making full use of the structural information.

## H  RELATED WORK

In order to quickly and accurately identify cell types, a series of cell annotation models have been developed in recent years, including reference-based and marker-based methods. Reference-based methods (Kiselev et al., 2018; Aran et al., 2019; Jia et al., 2023; Xu et al., 2023) utilize a reference dataset to train a classifier and then migrate it to a new dataset, which have achieved excellent performance on cell annotation tasks. However, such approaches require that the reference dataset and the target dataset are similar to each other, which may cause problems for scRNA-seq studies (Ianevski et al., 2022; Jia et al., 2023). Moreover, previous work (Lee et al., 2023) has suggest that marker-based methods require only marker information and are easier to use. Based on these considerations, this study focuses on marker-based methods.

Some marker-based approaches focus on improving heuristic association scoring strategies, such as the Cell-type-specific scoring method mentioned above and the Miko scoring method that can be adapted to different gene set sizes (Mikolajewicz et al., 2022; Ianevski et al., 2022). Meanwhile, another line of research proposes a two-stage workflow. For instance, Garnett (Pliner et al., 2019) first uses a heuristic-based approach to obtain association scores between cells and types, and then generates pseudo-labels and trains a generalized linear model based on this. MarkerCount (Kim et al., 2022) uses a counting strategy to assess associations between cells and types, and then modifies previous predictions by clustering results. scSorter (Guo & Li, 2021) is a classical marker-based single-cell annotation model that alternately optimizes cluster assignments and centroids to assign cells to different predefined cell types. We discover that due to the large amount of noise in the heuristic relevance scores, previous approaches have to correct for it further, with Garnett leveraging the fitting power of machine learning algorithms and MarkerCount utilizing the information from the clusters. sICTA introduces a basic self-training framework, however it ignores the properties of single-cell datasets. Previous approaches are at a cursory stage of exploring how to better optimize the results of heuristic strategy annotation. Unlike previous work, we introduce a new self-training strategy, which greatly enhances the model's ability to self-optimize. Furthermore, we propose contextualized association scoring CAS to improve the compatibility of marker and datasets. Notably, for a more comprehensive comparison, we uses CAS to generate pseudo-labels to fine-tune a large single-cell model, i.e., Geneformer (Theodoris et al., 2023) as our baseline.

---

**Algorithm 1** Overall Network Training

---

**Input:** Single-cell expression data $X$ and cell type marker genes $\{m_j\}$; Cell embeddings $\mathbf{E_c}$ obtained from large single-cell models; Number of epochs for pre-training $e_p$ and self-training $e_s$.

**Output:** The association between cell and cell type $\hat{Y}$.

1: Construct cell-cell networks $G$ based on $\mathbf{E_c}$
2: Calculate the CAS scores by using Eqs. (2-8) as the initial $\tilde{Y}^{(0)}$
3: **for** $i = 1, ..., e_p$ **do**
4:     Infer GNN Embedding $h_g \leftarrow f_g(X, G)$
5:     Infer Transformer Embedding $h_t \leftarrow f_t(X)$
6:     Estimate $\hat{Y}$ by Eqs. (13-14)
7:     Compute $\mathcal{L}_{\text{PreTrain}}$ by Eq. (17)
8:     Update $f(\cdot), f_g(\cdot, \cdot), f_t(\cdot)$
9: Update pseudo-label $\tilde{Y}^{(1)} \leftarrow \hat{Y}^{(1)}$
10: **for** $i = 1, ..., e_s$ **do**
11:     Infer GNN Embedding $h_g \leftarrow f_g(X, G)$
12:     Infer Transformer Embedding $h_t \leftarrow f_t(X)$
13:     Estimate $\hat{Y}$ by Eqs. (13-14)
14:     Compute $\mathcal{L}_{\text{SelfTrain}}$ by Eq. (18)
15:     Update $\tilde{Y}, f(\cdot), f_g(\cdot, \cdot)$, and $f_t(\cdot)$
16: **return** $Y$

---

Table 2: Performance of different methods, where **Avg** represents the average of the performance over all single-cell datasets, **Bold** denotes the optimal result, and underline denotes the second best result. **Std** denotes the standard deviation of all single-cell datasets, and its smaller the better.

| Model | Avg↑ | Std↓ | Muraro | Stoeckius | Zheng | Tirosh | Puram | Zeisel | Dominguez-Conde | Yoshida |
|---|---|---|---|---|---|---|---|---|---|---|
| | | | | | **Macro-Precision** | | | | | |
| scSorter | 0.580 | 0.254 | 0.927 | 0.562 | 0.610 | 0.678 | 0.065 | 0.826 | 0.519 | 0.536 |
| Garnett | 0.516 | 0.190 | 0.554 | 0.648 | 0.064 | 0.744 | 0.540 | 0.769 | 0.341 | 0.470 |
| SCINA | 0.627 | 0.164 | 0.859 | 0.742 | 0.645 | 0.830 | 0.342 | 0.702 | 0.481 | 0.452 |
| scType | 0.574 | 0.128 | 0.863 | 0.617 | 0.452 | 0.615 | 0.521 | 0.647 | 0.455 | 0.442 |
| MarkerCount | 0.736 | **0.107** | 0.830 | 0.794 | 0.794 | 0.826 | 0.782 | 0.843 | 0.472 | **0.618** |
| Geneformer | 0.719 | 0.111 | 0.607 | 0.768 | 0.822 | 0.830 | 0.835 | 0.756 | 0.585 | 0.548 |
| sICTA | 0.807 | 0.133 | 0.942 | 0.949 | 0.828 | 0.831 | 0.798 | 0.852 | 0.572 | 0.547 |
| CSSTA | **0.852** | 0.116 | **0.958** | **0.976** | **0.839** | **0.856** | **0.889** | **0.943** | **0.754** | 0.604 |
| | | | | | **Macro-Recall** | | | | | |
| scSorter | 0.564 | 0.249 | 0.949 | 0.449 | 0.562 | 0.416 | 0.167 | 0.870 | 0.572 | 0.582 |
| Garnett | 0.398 | 0.195 | 0.471 | 0.421 | 0.092 | 0.349 | 0.294 | 0.824 | 0.356 | 0.591 |
| SCINA | 0.654 | 0.170 | 0.859 | 0.871 | 0.689 | 0.891 | 0.387 | 0.626 | 0.496 | 0.453 |
| scType | 0.747 | 0.110 | 0.915 | 0.884 | 0.656 | 0.904 | 0.667 | 0.698 | 0.629 | 0.607 |
| MarkerCount | 0.729 | 0.128 | 0.861 | 0.859 | 0.756 | 0.925 | 0.800 | 0.775 | 0.530 | 0.541 |
| Geneformer | 0.758 | 0.116 | 0.632 | **0.947** | 0.768 | 0.924 | 0.797 | 0.692 | 0.661 | 0.644 |
| sICTA | 0.822 | 0.106 | 0.968 | 0.907 | 0.748 | **0.928** | 0.802 | 0.848 | 0.694 | 0.632 |
| CSSTA | **0.858** | **0.103** | **0.971** | 0.902 | **0.778** | 0.895 | **0.937** | **0.965** | **0.717** | **0.702** |

Table 3: Prediction accuracy of MarkerCount, sICTA, and CSSTA for the rarest cell type in each dataset. Percentages indicate the proportion of the rarest cell type in each dataset.

| Model | Avg (2.00%) | Muraro (3.80%) | Stoeckius (1.20%) | Zheng (3.20%) | Tirosh (1.60%) | Puram (1.60%) | Zeisel (3.30%) | Dominguez-Conde (0.80%) | Yoshida (0.50%) |
|---|---|---|---|---|---|---|---|---|---|
| MarkerCount | 0.433 | 0 | 0.295 | 0.418 | 0.913 | 0 | 0.806 | 0.099 | 0.161 |
| sICTA | 0.713 | **0.988** | **0.841** | **0.486** | **1** | 0 | 0.939 | 0.756 | 0.339 |
| CSSTA | **0.778** | 0.975 | 0.636 | 0.436 | 0.609 | **0.961** | **1** | **0.864** | **0.746** |

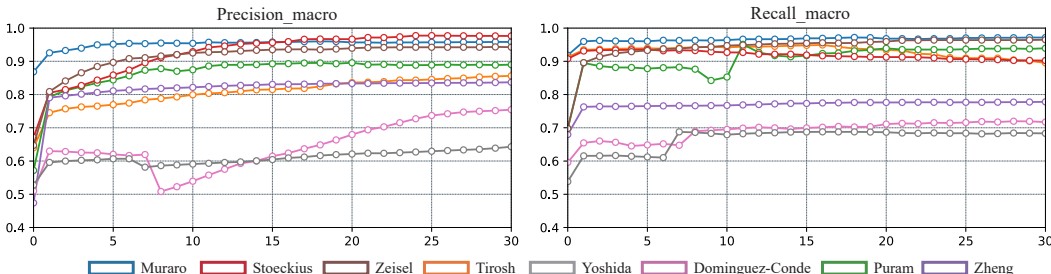

Figure 8: Plot of performance variation of CSSTA in self-training.

Table 4: Prediction accuracy of MarkerCount, sICTA, and CSSTA for the cell type with the highest percentage in each dataset. Percentages indicate the proportion of the rarest cell type in each dataset.

| Model | Avg (44.8%) | Muraro (38.7%) | Stoeckius (44.3%) | Zheng (71.2%) | Tirosh (73.0%) | Puram (44.7%) | Zeisel (31.2%) | Dominguez-Conde (27.1%) | Yoshida (28.0%) |
|---|---|---|---|---|---|---|---|---|---|
| MarkerCount | 0.803 | 0.979 | 0.994 | **0.942** | 0.831 | 0.918 | **0.973** | 0.064 | **0.726** |
| sICTA | 0.818 | 0.970 | 0.999 | 0.906 | 0.577 | 0.992 | 0.892 | 0.697 | 0.511 |
| CSSTA | **0.883** | **0.995** | 1 | 0.927 | **0.937** | **0.993** | 0.965 | **0.707** | 0.542 |

Table 5: The influence of different strategies on model performance during pseudo-label generation and supervision.

| Model | Avg↑ | Std↓ | Muraro | Stoeckius | Zheng | Tirosh | Puram | Zeisel | Dominguez-Conde | Yoshida |
|---|---|---|---|---|---|---|---|---|---|---|
| | | | | **Macro-Precision** | | | | | | |
| CSSTA w GT | 0.670 | 0.243 | 0.614 | 0.836 | 0.680 | 0.825 | 0.879 | **0.948** | 0.244 | 0.331 |
| CSSTA w AC | 0.740 | 0.173 | **0.967** | 0.948 | 0.529 | 0.843 | 0.787 | 0.779 | 0.521 | 0.544 |
| CSSTA w/o LPL | 0.751 | **0.105** | 0.900 | 0.814 | 0.833 | 0.716 | 0.712 | 0.828 | 0.577 | 0.626 |
| CSSTA w/o CAS | 0.793 | 0.147 | 0.962 | 0.963 | 0.810 | 0.853 | 0.641 | 0.895 | 0.535 | **0.681** |
| CSSTA | **0.852** | 0.116 | 0.958 | **0.976** | **0.839** | **0.856** | **0.889** | 0.943 | **0.754** | 0.604 |
| | | | | **Macro-Recall** | | | | | | |
| CSSTA w GT | 0.704 | 0.255 | 0.700 | 0.781 | 0.737 | 0.936 | 0.926 | **0.967** | 0.288 | 0.293 |
| CSSTA w AC | 0.738 | 0.170 | **0.979** | 0.889 | 0.711 | **0.944** | 0.705 | 0.638 | 0.513 | 0.527 |
| CSSTA w/o LPL | 0.802 | 0.126 | 0.945 | **0.941** | 0.773 | 0.920 | 0.709 | 0.872 | 0.638 | 0.622 |
| CSSTA w/o CAS | 0.767 | 0.119 | 0.978 | 0.894 | 0.728 | 0.857 | 0.706 | 0.691 | 0.680 | 0.605 |
| CSSTA | **0.858** | **0.103** | 0.971 | 0.902 | **0.778** | 0.895 | **0.937** | 0.965 | **0.717** | **0.702** |
| | | | | **Macro-F1** | | | | | | |
| CSSTA w GT | 0.640 | 0.258 | 0.649 | 0.692 | 0.630 | 0.828 | 0.892 | **0.957** | 0.210 | 0.258 |
| CSSTA w AC | 0.690 | 0.192 | **0.972** | 0.916 | 0.570 | 0.831 | 0.730 | 0.620 | 0.438 | 0.446 |
| CSSTA w/o LPL | 0.723 | 0.135 | 0.909 | 0.852 | 0.759 | 0.710 | 0.671 | 0.832 | 0.535 | 0.515 |
| CSSTA w/o CAS | 0.729 | 0.165 | 0.969 | 0.924 | 0.739 | 0.853 | 0.580 | 0.712 | 0.490 | 0.561 |
| CSSTA | **0.832** | **0.129** | 0.964 | **0.932** | **0.780** | **0.863** | **0.902** | 0.952 | **0.672** | **0.593** |
| | | | | **Micro-F1** | | | | | | |
| CSSTA w GT | 0.722 | 0.224 | 0.834 | 0.726 | 0.703 | 0.857 | 0.956 | **0.967** | 0.343 | 0.388 |
| CSSTA w AC | 0.754 | 0.157 | **0.979** | 0.965 | 0.663 | 0.830 | 0.773 | 0.730 | 0.572 | 0.522 |
| CSSTA w/o LPL | 0.764 | 0.157 | 0.946 | 0.957 | 0.870 | 0.698 | 0.675 | 0.870 | 0.547 | 0.550 |
| CSSTA w/o CAS | 0.805 | 0.150 | **0.979** | 0.965 | 0.844 | **0.960** | 0.660 | 0.783 | 0.546 | **0.700** |
| CSSTA | **0.895** | **0.107** | 0.975 | **0.971** | **0.886** | 0.948 | **0.973** | 0.964 | **0.777** | 0.667 |

Table 6: CAS scoring sensitivity to incomplete marker genes (x% = masking probability), with standard deviations across five trials shown in parentheses.

| Dataset | CAS | CAS 10% | CAS 20% | CAS 30% |
|---|---|---|---|---|
| | | **Macro-F1** | | |
| Tirosh | 0.644 | 0.637 (0.003) | 0.630 (0.004) | 0.621 (0.010) |
| Zheng | 0.510 | 0.505 (0.006) | 0.502 (0.013) | 0.492 (0.013) |
| | | **Micro-F1** | | |
| Tirosh | 0.619 | 0.615 (0.006) | 0.629 (0.012) | 0.615 (0.014) |
| Zheng | 0.591 | 0.588 (0.006) | 0.567 (0.016) | 0.569 (0.015) |

Table 7: CSSTA sensitivity to incomplete marker gene data.

| Dataset | CSSTA | CSSTA (CAS 10%) | CSSTA (CAS 20%) | CSSTA (CAS 30%) |
|---------|-------|-----------------|-----------------|-----------------|
| **Macro-F1** | | | | |
| **Tirosh** | 0.863 | 0.856 (0.006) | 0.865 (0.005) | 0.829 (0.033) |
| **Zheng** | 0.780 | 0.765 (0.006) | 0.749 (0.017) | 0.668 (0.072) |
| **Micro-F1** | | | | |
| **Tirosh** | 0.948 | 0.940 (0.018) | 0.956 (0.015) | 0.856 (0.132) |
| **Zheng** | 0.886 | 0.872 (0.003) | 0.854 (0.015) | 0.777 (0.052) |

Table 8: Sensitivity of model performance to hyperparameter $\tau$.

| Model | CSSTA ($\tau = 0.02$) | CSSTA ($\tau = 0.05$) | CSSTA ($\tau = 0.1$) | CSSTA ($\tau = 0.2$) |
|-------|------------------------|------------------------|-----------------------|-----------------------|
| **Macro-F1** | | | | |
| **Tirosh** | 0.865 | 0.863 | 0.868 | 0.813 |
| **Zheng** | 0.755 | 0.78 | 0.779 | 0.776 |
| **Micro-F1** | | | | |
| **Tirosh** | 0.954 | 0.948 | 0.945 | 0.771 |
| **Zheng** | 0.863 | 0.886 | 0.884 | 0.875 |

Table 9: Sensitivity of model performance to hyperparameter $\epsilon$.

| Model | CSSTA ($\epsilon\% = 10\%$) | CSSTA ($\epsilon\% = 20\%$) | CSSTA ($\epsilon\% = 30\%$) | CSSTA ($\epsilon\% = 40\%$) |
|-------|------------------------------|------------------------------|------------------------------|------------------------------|
| **Macro-F1** | | | | |
| **Tirosh** | 0.853 | 0.863 | 0.788 | 0.814 |
| **Zheng** | 0.773 | 0.780 | 0.734 | 0.715 |
| **Micro-F1** | | | | |
| **Tirosh** | 0.963 | 0.948 | 0.637 | 0.744 |
| **Zheng** | 0.874 | 0.886 | 0.847 | 0.825 |

Table 10: Performance of CSSTA combined with different foundation models. Genformer, scGPT, and CellPLM are fine-tuned using pseudo labels obtained through CAS, and then the final predictions are obtained. CSSTA (Genformer), CSSTA (scGPT), and CSSTA (CellPLM) represent the performance of CSSTA when utilizing embeddings from different single-cell foundational models.

| Dataset | Zheng | Zheng | Tirosh | Tirosh |
|---------|-------|-------|--------|--------|
|  | Macro-F1 | Micro-F1 | Macro-F1 | Micro-F1 |
| **CSSTA (Genformer)** | 0.78 | 0.886 | 0.863 | 0.948 |
| **CSSTA (scGPT)** | 0.778 | 0.884 | 0.876 | 0.971 |
| **CSSTA (CellPLM)** | 0.772 | 0.89 | 0.872 | 0.971 |
| **Genformer** | 0.772 | 0.877 | 0.795 | 0.682 |
| **scGPT** | 0.623 | 0.53 | 0.656 | 0.786 |
| **CellPLM** | 0.513 | 0.691 | 0.644 | 0.913 |

Table 11: Performance of different methods on single-cell datasets with hierarchical structure.

| Model | Macro-Precision | Macro-Recall | Macro-F1 | Micro-F1 |
|-------|-----------------|--------------|----------|----------|
| CSSTA (major) | 0.944 | 0.830 | 0.837 | 0.963 |
| CSSTA (minor) | 0.851 | 0.828 | 0.778 | 0.884 |
| HiCAT (major) | 0.667 | 0.784 | 0.692 | 0.968 |
| HiCAT (minor) | 0.799 | 0.795 | 0.793 | 0.843 |
| h_CSSTA (major) | **0.968** | **0.867** | **0.885** | **0.971** |
| h_CSSTA (minor) | **0.887** | **0.875** | **0.852** | **0.924** |

Table 12: Statistical information for the single-cell dataset. Ratio indicates the ratio of the type that accounts for the most cells and the number of cells contained in the rarest cell type.

| Dataset | Tissue | #Cell | #Class | Ratio | Protocol | Accession ID | Reference |
|---|---|---|---|---|---|---|---|
| **Muraro** | Human pancreas | 2,098 | 7 | 10.2 | CEL-Seq2 | GSE84133 | Muraro et al. (2016) |
| **Stoeckius** | Human blood | 7,467 | 6 | 36.9 | Drop-seq | GSE100866 | Stoeckius et al. (2017) |
| **Zheng** | Mouse blood | 68,302 | 6 | 23.0 | 10X genomics | GSE93421 | Zheng et al. (2017) |
| **Tirosh** | Human tumor (Melanoma) | 2,949 | 6 | 45.6 | Smart-Seq2 | GSE72056 | Tirosh et al. (2016) |
| **Puram** | Human tumor (HeadNeck) | 3,224 | 6 | 2.8 | Smart-Seq2 | GSE103322 | Puram et al. (2017) |
| **Zeisel** | Mouse brain | 3,005 | 7 | 9.5 | STRT-Seq | GSE60361 | Zeisel et al. (2015) |
| **Dominguez-Conde** | Human blood | 25,362 | 9 | 33.9 | 10X genomics | E-MTAB-11536 | Domínguez-Conde et al. (2022) |
| **Yoshida** | Human blood | 43,468 | 14 | 56.0 | 10X genomics | GSE168215 | Yoshida et al. (2022) |