# OpenReview forum: "Context-aware self-training framework for cell type annotation using marker genes"
_ICLR.cc/2026/Conference — ICLR 2026 Conference Withdrawn Submission_

### Official Review · Reviewer_tpZm · 2025-10-28

**Soundness:** 3
**Presentation:** 2
**Contribution:** 2
**Rating:** 2
**Confidence:** 4

**Summary:**

The paper aims at tackling the cell-type annotation problem by solving two problems: how to best initialize pseudo-labels in single cell datasets, and how to leverage these labels using self-supervise learning to improve the cell-annotation. Specifically, the paper proposes CSSTA, a context-aware self-training framework for marker-based single-cell RNA-seq annotation. It first builds “contextualized association scoring” (CAS), which improves upon classic marker specificity leveraging dataset-dependent statistics to generate higher-quality pseudo-labels. It then separates high- vs low-confidence pseudo-labels with type-adaptive thresholds and supervises them with different losses.

Across eight benchmark datasets, CSSTA outperforms marker-based and self-training baselines. The authors also show CSSTA convergent self-training behavior, as well as thorough evaluation of model's both branches through ablation studies. Finally, they provide an extension (h_CSSTA) that improves hierarchical annotation.

**Strengths:**

The authors proposed a novel model to improve current self-supervise learning models for cell annotation by proposing an improved way of pseudo-annotating the cells. The results are quite robust (the evaluate their model on several datasets with consistent good results across multiple metrics), and provides appropriate ablation studies.

The work also provides a nice natural hierarchical extension with gains over previous models such as HiCAT.

**Weaknesses:**

The paper's notation is very hard to follow. there is notation that it is used before being defined (e.g., 'y', or notation that is heavily abused (such as 'n' or 'm'). Also please do not use superscript for indexing (unless it is in brackets) since it can be confused with exponentiation. Example: Equation (3) is very difficult to interpret. In summary, the section 2 is very confusing making it very difficult to evaluate the novelty and relevance of the proposed model.

Importantly, the state-of-the-art review lacks both recent and highly used cell type annotators (e.g., Seurat label transfer/JIND for label transfer, scPred/CellTypist for automatic annotators, UCell/AUCell for signature scoring). As well as a description of the different modalities (trained, one-shot, etc.).

Minor:

The text says the GNN “captures the spatial relationships between cells,” which is potentially misleading in scRNA-seq (it models transcriptional-similarity graphs, not spatial transcriptomics)

Graph construction fixes a 1% edge threshold without a reported sensitivity study

**Questions:**

The set T_i is not clear how it is defined. If in eq. (5) how come it sums over the cells belonging to type j if I don't have the annotation yet? not sure how this can be computed.

Why using the Relu function if it is also unbounded?

How does the model compare with current gold-standard annotators for single cell genomics?

---

### Official Review · Reviewer_7kK6 · 2025-10-30

**Soundness:** 3
**Presentation:** 3
**Contribution:** 1
**Rating:** 2
**Confidence:** 4

**Summary:**

The authors introduce CSSTA, a self-training model for cell type annotation of single-cell RNA-seq data using marker genes. CSSTA is an extension of sICTA that utilizes a different marker gene scoring method to generate pseudo labels, employs a distinct loss function by using a per-cell-type confidence threshold, and incorporates a pre-trained transformer and a GNN to extract cell-cell level information.

**Strengths:**

- The paper provides a good ablation study that shows the contribution of individual components to the overall performance.
- The figures generally illustrate the methodology well.
- On the evaluated datasets, the methods seem to perform well.
- The authors propose a method to address a common issue when analyzing single-cell RNA-seq datasets.
- The code is publicly available with good documentation.

**Weaknesses:**

**Major:**
- The paper's contribution is significantly limited by substantial overlap with existing work [1].
- Table 1 does not present the standard deviation on performance estimates, and given the significant deviation in Figure 6 it might be that CSSTA does not significantly outperform existing methods.
- The datasets used for evaluation are relatively small and mostly outdated. Modern single-cell datasets can contain millions of cells, and evaluation on larger, more recent datasets would show its utility.
- Figure 4 reveals no significant performance difference between cell-type-specific scoring and CAS.
- The paper omits comparisons with probably the most widely used marker gene scoring methods, Scanpy's score_genes functions [2].
- The paper lacks any runtime comparison with existing methods, making it difficult to assess the trade-off between performance gains and computational cost.

**Minor:**
- Several captions are too short and should include more information about the figures. E.g. "Figure 4: Average performance of different association scoring strategies." On which dataset(s) is this analysis done? What do the error bars represent?
- The y-axis labels and legend in Figure 4 are too small.
- Figure 1a resembles a fully-connected neural network but should represent sparse gene-to-cell-type associations, as marker genes are typically specific to particular cell types.
- The x-axis label for Figure 8 is missing.

[1] Hegang Chen, Yuyin Lu, Yanghui Rao, A self-training interpretable cell type annotation framework using specific marker gene, Bioinformatics, Volume 40, Issue 10, October 2024, btae569, https://doi.org/10.1093/bioinformatics/btae569

[2] Wolf, F., Angerer, P. & Theis, F. SCANPY: large-scale single-cell gene expression data analysis. Genome Biol 19, 15 (2018). https://doi.org/10.1186/s13059-017-1382-0

**Questions:**

- What is the standard deviation across different runs for the estimates shown in Table 1?
- How does the method compare on larger datasets in terms of performance and run time to established methods?
- Why was Scanpy's score_genes function not included as a baseline for associating marker gene expression with cell types?
- Line 230 states, "Since the average frequency is unbounded, we scale it with the ReLU function." ReLU is linear for positive values and thus does not bound the output. What is meant here?
- In Table 1, does "Std denotes the standard deviation of all single-cell datasets" refer to the standard deviation of all performance metrics across all datasets?
- Figure 8 shows the number of epochs?

---

### Official Review · Reviewer_ZHPV · 2025-11-05

**Soundness:** 2
**Presentation:** 2
**Contribution:** 2
**Rating:** 4
**Confidence:** 4

**Summary:**

The paper provides a new method for single-cell annotation. Based on the new self-training, GNN, and attention, the model achieves better annotation. The self-training method is from the observation of context information and confidence. The method also constructs the graph of cell-cell association to aid the learning process. The benchmark experiments show an increase in the evaluation.

**Strengths:**

1. The research field and the application of the method are important.
2. The new observation in the field and the findings from the methods could be extended to improve the other methods working in this field.

**Weaknesses:**

1. The experiment should provide more results and analysis to support the claim in the introduction and design of the methods, instead of only showing the increase in the number for the whole parts.
2. The key design of the method contains self-training and a Graph learning module. The self-training needs further verification. The method is not significant enough. And the ablation study shows that the attention module provides the key contribution in the design, which raises concerns about the effectiveness of the method
3. The author should provide a detailed explanation of the "context" in the paper. It is hard to understand the word unless you finish reading the whole paper. It is not the context in LLM, and also the general context in bio. It provides a specific meaning in the paper and could help the readers to get the key observation and the motivation of the design.

**Questions:**

1. Could authors please provide the details about the claims in the introduction in the datasets, and how and to what extent the methods solve them through experiments?
2. Since the graph is constructed in the method given by the definition, could the authors provide some potential bias or the information loss in this construction, if any?

---

### Official Review · Reviewer_JPPJ · 2025-11-07

**Soundness:** 3
**Presentation:** 3
**Contribution:** 2
**Rating:** 6
**Confidence:** 3

**Summary:**

A self-training framework for annotating scRNA-seq data is proposed which leverages marker genes to get pseudo labels, where only high confidence cells are used and then uses self-training to annotate the entire dataset. Authors propose using a dual view architecture where transformer and GNN branches are used to get one single embedding for each cell. Prototypical learning is used to classify the cells. Results demonstrate improved performance on multiple datasets.

**Strengths:**

- Results are impressive, the performance of the proposed method is better than all compared approaches.
- Using cell-type specific thresholds makes sense because some types might be easier to classify than others
- Using marker genes to inform clustering is novel, almost like a few shot learning problem. I would preferred to see some parallels to that in the related work or discussion.

**Weaknesses:**

- Paper is very hard to read, some of the details are in supplementary which are important for example, pretraining loss and self training losses are different. This is important because without pre-training, the embeddings might be wrong completely (no training). Similarly prototypical learning could be moved to supplementary since it is not foundational to the idea.
- Authors only compare with methods from 2024. Please add comparison to scTrans (Zou et al. 2025, Plos). Similarly compare with methods presented in that paper too.
- Novelty is limited (mainly the architecture but is overemphasized by authors) because the proposed approach just combines multiple ideas. Using cell-type specific threshold has been proposed earlier in JIND (Goyal et al. 2022). Learning from pseudo labels using high confidence cell-type specific threshold is not novel and obvious.
- I have several questions about the approaches, please look at the questions section.

**Questions:**

- Authors use a single cell foundation model which is pretrained, do other methods use this foundation model as well?
- The method seems to be very sensitive to the initial pseudo labels which assumes access to all cell-types and marker genes. What about rare cell-types, is there any way to reject them or what if the list is missing a cell-type?
- How did the authors tune the hyper parameters for their approach, is there any way to validate if the method works on a new unseen dataset? Were there any criteria used to stop pretraining at k epochs or self training (such as convergence). How about pseudo labeling thresholds?

---

### Official Review · Reviewer_2sKm · 2025-11-07

**Soundness:** 2
**Presentation:** 2
**Contribution:** 2
**Rating:** 4
**Confidence:** 3

**Summary:**

This paper proposes CSSTA, a semi-supervised framework that iteratively refines pseudo-labels for single-cell annotation. The method formulates the task as an iterative optimization between feature representation learning and pseudo-label updating. The approach introduces a stability-aware self-training scheme with marker-gene priors and z-score-based feature normalization. Experiments on several public scRNA-seq datasets demonstrate improved cell-type prediction accuracy compared with a few baselines.

**Strengths:**

- A clear formulation of the pseudo-label refinement process for single-cell annotation, bridging semi-supervised learning and domain knowledge (marker genes).

- Demonstrates competitive performance on multiple datasets.

- The paper identifies and addresses a relevant challenge in single-cell data—label scarcity—and offers a practical method for leveraging unlabeled data.

**Weaknesses:**

- Convergence and stability: The method converts label refinement into an iterative pseudo-label optimization, but the paper lacks theoretical or empirical evidence about the convergence or stability of this process. Some stability analyses (e.g., monitoring pseudo-label consistency or loss oscillation) would make the approach more convincing.

- Notation conflict: Furthermore, the multiple uses of R (in Eqns 1, 3, 8) appear to denote different quantities but share the same symbol, which can be confusing.

- Baseline coverage: The baselines could be strengthened. Classical probabilistic methods such as scANVI are strong semi-supervised benchmarks. For pretraining-based models, scGPT [1], scFoundation [2], and scCello [3] are widely recognized and relevant comparisons, particularly since scCello also incorporates relational priors.

[1] scGPT: toward building a foundation model for single-cell multi-omics using generative AI
[2] Large-scale foundation model on single-cell transcriptomics
[3] Cell-ontology guided transcriptome foundation model

**Questions:**

1. In Eqn (3), why applying z-score transformation here but not in other scores?

2. The threshold in Eqn (9) seems manually chosen, but no ablation or sensitivity analysis is provided. Showing how varying this threshold affects prediction accuracy would improve interpretability.

3. Although the paper mentions class imbalance, it does not evaluate how CSSTA performs on minor cell types. Reporting per-class F1 or macro-averaged metrics would be valuable.

4. The framework assumes reasonably complete marker-gene knowledge. It would be important to test performance when marker information is noisy or partially missing, reflecting realistic biological settings.

5. Maybe adding more baselines as stated in Weakness.

6. As stated in Weakness, is there any theoretical or empirical guarantee for the convergence or stability of the pseudo-label iteration process?

---

### Note · Authors · 2025-11-24

I have read and agree with the venue's withdrawal policy on behalf of myself and my co-authors.